# Are Danes' Immigration Policy Preferences Based on Accurate Stereotypes?

**Emil O. W. Kirkegaard** [1,*] **, Noah Carl** [2] **and Julius D. Bjerrekær** [3]

1    Ulster Institute for Social Research, London NW26 9LQ, UK
2    Independent Researcher, Cambridge, UK; noah_carl3742@hotmail.com
3    Independent Researcher, Austin, TX, USA
*    Correspondence: emil@emilkirkegaard.dk

**Abstract:** Stereotypes about 32 country-of-origin groups were measured using an online survey of the adult, non-elderly Danish population (n = 476 after quality control). Participants were asked to estimate each group's net fiscal contribution in Denmark. These estimates were then compared to the actual net fiscal contributions for the 32 groups, taken from a report by the Danish Ministry of Finance. Stereotypes were found to be highly accurate, both at the aggregate level (r = 0.81) and at the individual level (median r = 0.62). Interestingly, participants over- rather than underestimated the net fiscal contributions of groups from countries with a higher percentage of Muslims. Indeed, this was true at both the aggregate and individual levels (r = −0.25 and median r = −0.49, respectively). Participants were also asked to say how many immigrants from each group should be admitted to Denmark. There was a very strong correlation between participants' aggregate immigration policy preferences and their estimates of the 32 groups' fiscal contributions (r = 0.98), suggesting that their preferences partly reflect accurate stereotypes. Most of the analyses were pre-registered.

**Keywords:** stereotypes; stereotype accuracy; immigration; policy preferences; Denmark

## 1. Introduction

Since the beginning of the European migrant crisis, researchers have become increasingly interested in how different characteristics of prospective immigrants affect Europeans' immigration policy preferences [1–4]. In a recent study, Carl [5] examined the relationship between British people's immigration policy preferences for 23 origin countries and the arrest rates of immigrants from those countries living in the UK. He found that net opposition to immigrants from the 23 origin countries was strongly related to their arrest rates, r = 0.69 [0.39, 0.86]. One possible interpretation of his result is that people rely, at least in part, on unmeasured but accurate stereotypes about origin country groups to inform their immigration policy preferences. This interpretation is consistent with a recent study by Kirkegaard and Bjerrekær [6], who found that Danish people's estimates of welfare use rates for 70 origin country groups correlated at r = 0.70 [0.55, 0.80] with the actual use rates for those groups. Both of the preceding findings are in line with a larger body of research on stereotype accuracy for demographic groups [7–10]. (Note that both Carl's [5] and Kirkegaard & Bjerrekær's [6] effect sizes correspond to consensual stereotype accuracy correlations [9].)

An alternative explanation for the results obtained by Carl [5] and Kirkegaard & Bjerrekær [6] is that stereotypes held by the host population cause differential outcomes across origin country groups via the process of self-fulfilling prophecy [11–14]. In other words, it is possible that all origin country groups start out with roughly the same average characteristics (e.g., same arrest rate, same use of social benefits), but that stereotypes held by the host population set in motion a process of a self-fulfilling

prophecy, leading origin country groups to gradually diverge from one another. While this alternative explanation should not be dismissed out of hand, there are a number of reasons to doubt its veracity.

First, it is unclear why members of the host population would come to hold particular stereotypes if those stereotypes did not in some way correspond with reality. One possibility is that Europeans would be naturally biased against, say, non-white groups or non-Christian groups. However, the fact that some non-white, non-Christian groups often have among the most favorable outcomes suggests that this is unlikely to be the case (see [15]). For example, Japanese living in the UK had the lowest average arrest rate in Carl's [5] study, while Indians living in Denmark had among the lowest average welfare use rates in Kirkegaard & Bjerrekær's [6] study. Moreover, there is evidence that differential immigrant outcomes can often be explained, at least in part, by skills present upon arrival. In other words, origin country groups with better education tend to have more favorable outcomes in their host country [16–18]. For example, Carl [18] reported a correlation of r = 0.56 between a skill-selectivity contrast (percentage high-skilled minus percentage low-skilled) and log median household income across 75 origin country groups in the United States. Given that the vast majority of immigrants arrive after completing their education, the host population's stereotypes cannot have caused the observed differences in education levels across origin country groups. A plausible causal pathway is therefore from skills present upon arrival, to differential immigrant outcomes, to the host population's stereotypes.

Second, Jussim [8,19] reviewed half a century of literature on whether self-fulfilling prophecies cause stereotype accuracy and concluded that "although errors, biases, and self-fulfilling prophecies in person perception are real, reliable, and occasionally quite powerful, on average, they tend to be weak, fragile, and fleeting". For example, expectancy effects on student achievement are typically around r = 0.20, despite the fact that teacher expectations of student achievement correlate with actual achievement at r = 0.40 to r = 0.80. In addition, such effects are often larger earlier in the teaching year, when teachers are less well-acquainted with their students. Note that the classroom is a setting where there is direct and consistent interpersonal contact between the stereotype holder (i.e., the teacher) and those whose outcomes he or she is putatively affecting (i.e., the students). Hence one might expect that expectancy effects would be larger here than in many other contexts. Furthermore, Jussim [8,19] points out that a number of the most influential and highly cited studies in the literature on self-fulfilling prophecies were afflicted by serious methodological limitations or have proven difficult to replicate [20,21].

A study by the Danish ministry of finance reported net fiscal contributions for 32 origin country groups in Denmark [22]. There were very large differences between some of the groups (see Supplementary Materials). For example, the net contributions of British immigrants was about 12,000 Euros/year per person, while the net contribution of German immigrants was about 3600 Euros/year per person. By contrast, the net contribution of Turkish immigrants was −6300 Euros/year per person, and the net contribution of Syrian immigrants was −40,000 (Euros/year per person). Given the prominence of migrants' socioeconomic outcomes in the scholarly literature and public debate surrounding immigration [1,4], these estimates provide an interesting way to investigate stereotype accuracy about different origin country groups, as well as how stereotypes relate to people's immigration policy preferences.

The primary aims of the present study were twofold. First, we sought to replicate an earlier study of stereotypes about origin country groups in Denmark using slightly different measures for the stereotypes and the actual outcome. Second, we sought to examine the relationships between stereotypes, actual outcomes, and immigration policy preferences for the same origin country groups. Our key hypotheses were that: respondents would hold accurate stereotypes about origin country groups' net fiscal contributions; that they would display greater opposition to origin country groups with more negative net fiscal contributions in Denmark; and that the association between net fiscal contributions and immigration policy preferences would be mediated by respondents' stereotypes about origin country groups' net fiscal contributions.

## 2. Data and Questionnaire Design

Like Kirkegaard & Bjerrekær [6], we used an online pollster (Survee) to recruit a sample of participants for the study. This sample was taken from a pool of survey respondents that aimed to be approximately representative of the non-elderly, adult Danish population (ages 18–65) with respect to age, sex, educational attainment and location. The mean age in our sample was 39.3 (SD = 13), compared to an expected value of 41.4 (SD = 14) in the general population. The percentage male in our sample was 51.3%, compared to an expected value of 50.5% in the general population. Due to an oversight at the design stage, we did not collect data on respondents' educational attainment. However, the pollster was subsequently able to provide such data for around 60% of our respondents, based on archival sources. When we compared the distribution in our sample to that in the general population, we found that people with higher levels of education were somewhat over-represented in our sample. For example, the percentage without a high school degree was 9.1% in our sample, compared to 20.0% in the general population; the percentage with no more than a high school degree was 14.1% in our sample, compared to 11.3% in the general population; and the percentage with a university degree was 14.1% in our sample, compared to 11.5% in the general population.

When we compared the percentages of respondents identifying with each political party to the then-current poll shares for those parties (mean of 2 closest polls), we obtained a correlation of r = 0.94, suggesting good representativeness with respect to political preferences. Our goal was to recruit 500 persons. We continued sampling until we had approximately 500 persons (505, to be exact) who passed the first 5 quality control checks, which were simple attention questions asking the user to select a specific response option. Because 29 participants (of 505) who passed the first quality 5 checks failed the 6th check (i.e., not giving the same estimate for every origin country), and because of a minor error on the part of the pollster, the final sample was slightly lower than the desired 500. (The error of the pollster was to have allowed non-responses to questions for the first few responses, which led to unwanted missing data.) In total, we collected 850 responses of which 476 (56%) passed our stringent, pre-determined quality control. Data were gathered from 28 May to 19 July 2017. (The references figures for age, sex and education were taken from Statistics Denmark [23,24]. Those for age and sex correspond to the population aged 18–65, while those for education correspond to the population aged 15–69. The polls were taken from Wikipedia [25].)

### 2.1. Survey Design

We employed a randomized survey design with three separate sections. Randomization was used in order to examine whether the order of presentation had any effect (i.e., does it matter if we measure people's stereotypes before measuring their immigration policy preferences?). Having participants give their estimates of net fiscal contributions might cause them to change their policy preferences, if only temporarily. By randomizing the order of the sections, we split our sample into 6 groups because there are 6 ways to choose 3 from 3 options without repeats. Each option had an equal probability of being chosen and were therefore roughly evenly balanced in the final dataset (i.e., there was no/little differential order related drop-out). The full survey can be found in the Supplementary Materials (both original Danish and an English translation).

Section 1 measured participants' immigration policy preferences for the 32 origin countries. Specifically, participants were asked to say for each origin country whether there should be more, the same, fewer or no immigrants admitted to Denmark. This was the same as the measure utilized in Carl's [5] study, and was chosen to maximize comparability (the measure was originally taken from a YouGov poll). The 32 origin countries constitute every one of those included in the report by the Danish Ministry of Finance, from which our criterion data were taken [8,22]. These data correspond to the fiscal year 2014–2015.

Section 2 measured the estimates of each origin country group's net fiscal contribution. As the relevant scale (Euros per person per year) is not intuitive or well known, participants would likely have found it too difficult to estimate the values in real units. Indeed, the authors of the previous study

of origin country stereotypes had some trouble getting participants to understand how to rate different groups on the ratio scale utilized in that study [6]. For this reason, we used a simpler, 7-point Likert scale ranging from a 'very positive' to a 'very negative' net fiscal contribution. It is important to note that the stereotypes measured in our study correspond to a single criterion variable, namely net fiscal contribution, meaning that our study focusses on just one of the many domains in which migrants may influence their host societies. (We use the term 'stereotypes' as a shorthand for 'stereotypes about origin country groups' net fiscal contributions'.)

Section 3 measured a number of auxiliary variables. We did not need to measure age and sex as this information was supplied to us by the pollster. Instead, we measured participants' agreement with each major political party in Denmark, and the party they would vote for if there was an election tomorrow, as well as 3 questions about immigration:

1.	Overall, how are Muslims treated in Denmark in comparison to non-Muslims? 1–7 Likert scale from 'much better' to 'much worse'.
2.	On a scale of 1–7 (Likert agreement), how much do you agree with the following statement: Denmark should only allow immigrants who do not harm the public budget.
3.	On a scale of 1–7 (Likert agreement), how much do you agree with the following statement: Currently, non-Western immigrants pay on average more in tax than they receive in form of social benefits.

The first question was included because we wanted to examine the possibility that there might be a pro- or anti-Muslim bias in participants' estimates, as was done in the previous study mentioned above. The two latter questions were included due to their relevance to our outcome variable.

*2.2. Other Data*

We used data from a number of other sources.

The percentage of Muslims for each group was estimated using the home country's value. These estimates were taken from the database assembled by Pew Research for the year 2010 [26]. Using these estimates rests on the assumption that the immigrants from each country are roughly representative in terms of their religious beliefs and that their descendants are, too. Any self-selection with respect to religious attitudes will therefore introduce some error into our estimates. Yet previous research indicates that this issue is unlikely to be a major concern. For example, Koopman [27] examined a large 2008 survey of people living in 6 Western European countries (Germany, France, the Netherlands, Belgium, Austria and Sweden). He found that 97% of immigrants (and their children) in his sample from Morocco and Turkey identified as Muslim. By contrast, among members of the native population, 30% did not identify as Christian.

For a few ex-countries (e.g., former Yugoslavia), Pew Research did not provide any estimate for the percentage of Muslims in the population. In these cases, the percentage of Muslims was estimated by aggregating from the constituent countries (i.e., Serbia, Montenegro, etc. in the case of Yugoslavia).

Finally, we used data collected for the previous stereotype study in some secondary analyses [6]; the data are provided in the study's Supplemental Materials.

## 3. Analyses

Most of the analyses in our study were pre-registered before data collection began. In addition, for some of our analyses, specific numerical predictions (rather than mere directions of association) were made based on prior research and theory [28–30]. However, there were a couple of minor deviations from the pre-registered analyses: we employed an additional measure of anti-Muslim preference in Section 3.2.3, and we included a few exploratory comparisons using British data in Section 3.1.4. These additional analyses were not conceived at the time that data collection began.

### 3.1. Aggregate Stereotypes

Stereotype accuracy is commonly examined at both the individual (personal) level and the aggregate (consensual) level [9]. Each of these levels is interesting for different reasons. Aggregate stereotypes are theoretically the most important for social performance as they reflect the overall perception of a group, and are thus likely to influence the average or most prevalent behavior toward the group by other members of society (e.g., hiring practices). Individual stereotypes, on the other hand, provide the opportunity to examine person-level correlates of stereotype accuracy and bias. A relatively large body of research indicates that stereotypes are quite accurate at the aggregate level for most kinds of groups, especially major demographic groups [9,10,31]. Note that accuracies of r = 0.80 are not uncommon in this literature [32].

#### 3.1.1. Accuracy

As we only collected interval-level data for our stereotypes, we were unable to score accuracy using methods that require ratio-level data (e.g., the absolute distance to criterion value [5]). Instead, we relied upon Pearson's r as our measure of accuracy. This quantifies relative accuracy and takes into account relative distances, not just order (unlike Spearman's rho). Aggregate accuracy was thus quantified by first computing the mean estimates for each of the 32 origin countries, and then correlating these with the criterion values. Previous research has shown that other measures of central tendency do not result in superior accuracy [6]. Figure 1 displays the relationship between the aggregate estimates and the criterion values.

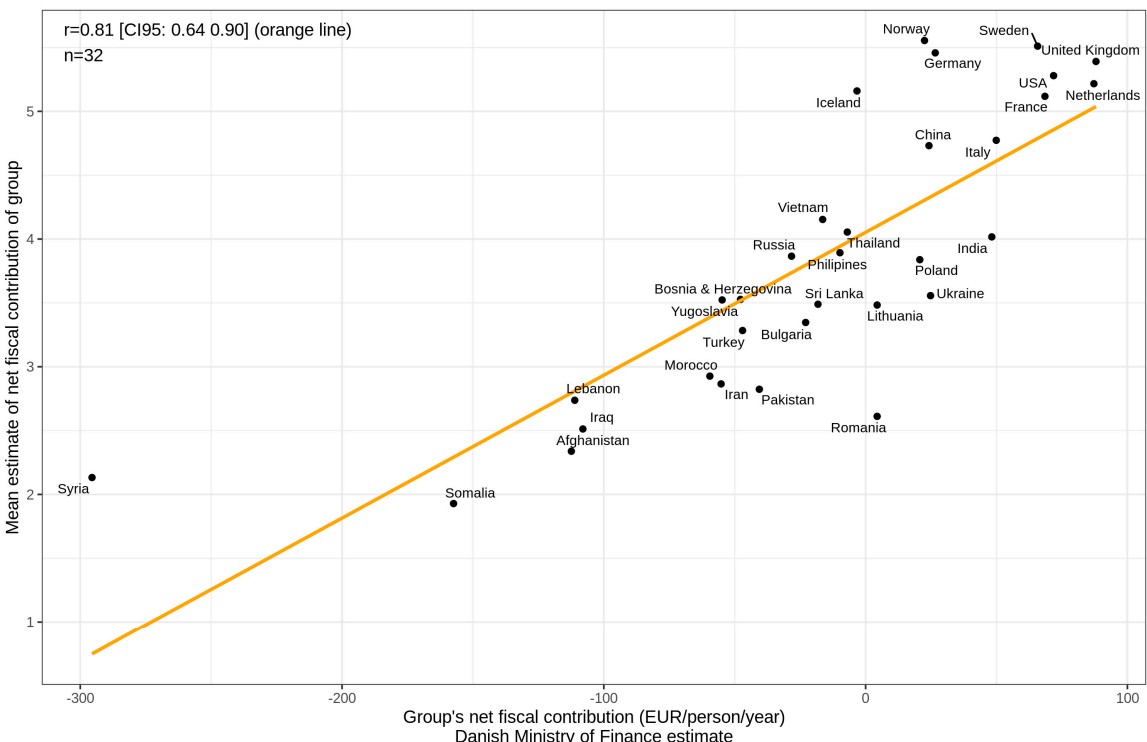

**Figure 1.** Aggregate stereotype accuracy (Pearson r). Y-axis values are group's estimated net fiscal contributions.

As in previous studies, the aggregate stereotypes were found to be highly accurate [9]. It is clear that Syria is a substantial outlier (as it also was for many of the following analyses). This is most likely because many migrants from this group had just arrived in the country as part of the exodus of Syrian refugees during the 2014–2015 European migrant crisis. Indeed, the number of Syrians in Denmark increased dramatically from ~8000 at the beginning of 2014 to ~14,100 by January 2015 [33]. Insofar as they had just arrived in the country, they would not have had full access to the labour market,

despite being eligible for state assistance (including subsidized housing, as well as a monetary stipend). Moreover, the vast majority of Syrians in Denmark were initially housed in special immigration centres, meaning that they would not have come into contact with members of the public. As a consequence, the reported net fiscal contribution for this group is arguably less representative of Syrians in Danish society than the other reported figures are for their corresponding groups. Excluding it from the present analysis increased accuracy from r = 0.81 to r = 0.85. In our pre-registration, we predicted a correlation of r = 0.84 (0.70, 0.92) based on the previous study. The current finding is therefore closely in line with our expectations, regardless of whether or not Syria is included (in the Supplementary Materials, we report all of our main results with Syria excluded). The overall correlation between results with and without Syria was r = 0.98, and the mean absolute delta was 0.029).

Furthermore, the aggregate stereotypes of the current study correlated at |r| = 0.94 with those in the previous study, despite the fact that participants were asked to estimate different outcomes (net fiscal contribution vs. percentage of persons aged 30–39 receiving social benefits). The correlation increased to |r| = 0.95 when excluding Syria. In the present context, aggregate stereotypes seem to have high reliability across related target traits/outcomes.

### 3.1.2. Muslim Bias in Stereotypes

Evidence indicates that many Muslim-majority communities in Denmark tend to have quite unfavorable socioeconomic outcomes (e.g., [6,15]). One possibility worth considering is that there are negative stereotypes about Muslims which hamper their ability to get ahead in Danish society (i.e., that there is a self-fulfilling prophecy at work with respect to outcomes like income, criminality and use of social benefits [8]).

The most plausible mechanism by which negative stereotypes might hamper the social performance of a particular group is other members of society underestimating the true social performance of that group. For example, if an important class of people such as employers consistently underestimate the job performance of a particular group, they will be less likely to recruit from that group than would be economically rational. However, there is little published research examining whether stereotypes about Muslims actually are negatively biased (i.e., whether people do systemically underestimate the social performance of Muslims). As far as the authors are aware, the only published study is the one mentioned above [6]. This study found that stereotypes were slightly 'biased' in favor of Muslims: people's estimates of Muslims' net fiscal contributions were slightly too high relative to criterion values, not too low. (We should note that our study does not seek to make any claims about the level of employment or other discrimination against Muslims in Denmark.)

There are multiple ways one can measure stereotype bias with respect to a particular outcome of interest. The previous study used two related metrics. The first ('Muslim error r') involved calculating the delta (difference) between each estimate and each criterion value and then correlating these with the Muslim percentage for each group. The second ('Muslim elevation error') involved calculating the weighted mean delta using both percentage Muslim and percentage non-Muslim as the weights and then subtracting the second from the first. This gives the Muslim-specific elevation error. In the previous study, these two metrics correlated at r = 0.96 across individuals, and so were essentially interchangeable.

The calculation of delta scores, however, requires that one has data on the same scale for both metrics, which the present study did not (estimates were interval-level in quasi-Likert units, whereas the criterion data were ratio-level in real units). As a consequence, two similar but alternative measures of stereotype bias were utilized. The first ('Muslim resid r') involved correlating the Muslim percentage for each group with the residual from a model of criterion values regressed on estimated values. The second involved weighting the mean standardized residual using the percentage Muslim for each group as the weights. To examine the comparability of the metrics, the data from the previous study were reanalyzed by calculating all metrics at the individual-level. Table 1 shows the results. Note that the second alternative metric turned out to be numerically equivalent to the first metric and thus was left out.

**Table 1.** Correlations among bias metrics. n = 484. Note that these analyses are based on data from Kirkegaard & Bjerrekær [6].

|  | **Muslim Resid r** | **Muslim Elevation Error** | **Muslim Error r** |
|---|---|---|---|
| Muslim resid r | 1 |  |  |
| Muslim elevation error | 0.71 | 1 |  |
| Muslim error r | 0.65 | 0.96 | 1 |

We see that the alternative metric had a moderately high correlation with the two metrics that require ratio-level data. Assuming that the new metric sufficiently captures the relevant phenomenon, we can calculate the Muslim bias for data collected in the present study, as shown in Figure 2. A correlation of r = −0.25 was observed at the aggregate level, indicating the presence of a slight pro-Muslim bias. Given the small sample (n = 32), the confidence interval (CI) spanned zero. The correlation increased to r = −0.30 when excluding Syria (see Supplementary Materials). Table 2 displays a comparison of Muslim biases for the aggregate data as a function of the method and dataset choices. While the effects are small and often have CIs that span zero (or nearly so), they are consistent in direction. These negative associations may be attributable to social desirability bias in respondents' answers. However, the fact that the survey was anonymous militates against this interpretation. Indeed, we cannot even be sure that respondents were cognizant of the percentage of Muslims in the home country when estimating net fiscal contributions, only that they did not show any tendency to underestimate the net fiscal contributions of groups for which the percentage was higher.

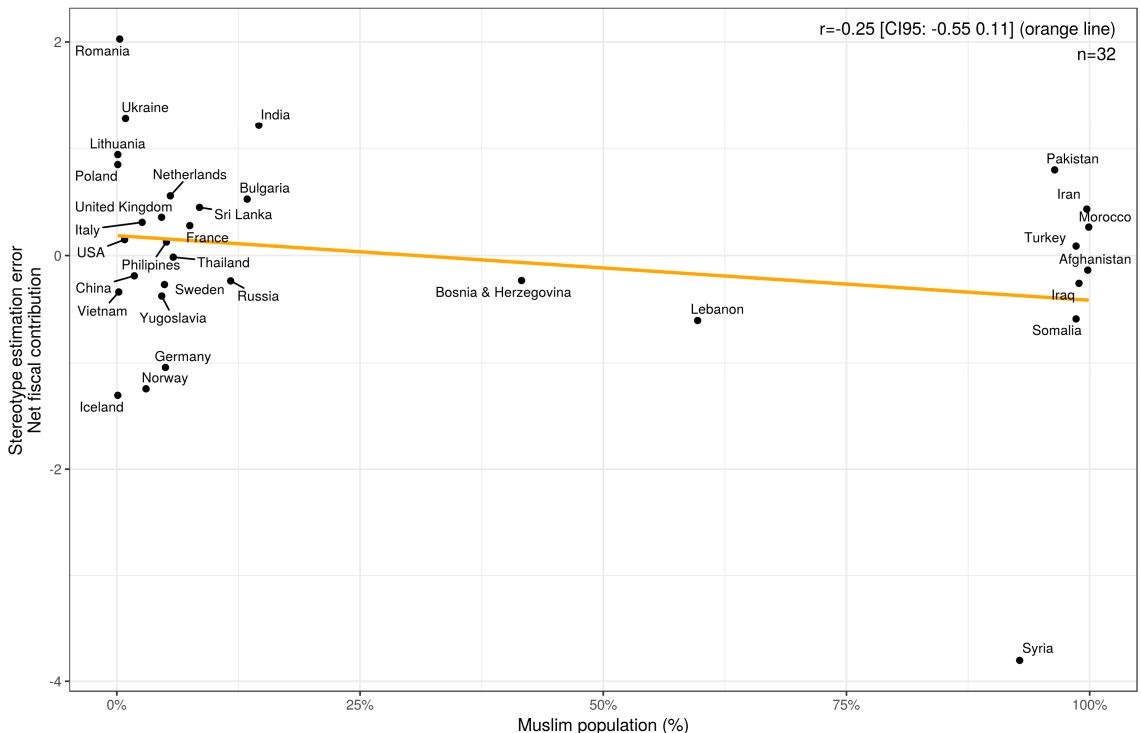

**Figure 2.** Muslim bias in aggregate stereotypes.

**Table 2.** Comparison of Muslim bias in aggregate stereotypes across methods and datasets. New data is from the present study. Old data is from Kirkegaard and Bjerrekær [6]. The calculations in the second and fourth rows (n = 32) were done in the sub-sample of the old data comprising the countries available in the new data.

| Data | n | Metric | Value |
|------|---|--------|-------|
| New | 32 | Muslim resid r | −0.25 [−0.55, 0.11] |
| Old | 32 | Muslim resid r | −0.11 [−0.44, 0.25] |
| Old | 70 | Muslim resid r | −0.27 [−0.48, −0.04] |
| Old | 32 | Muslim error r | −0.39 [−0.65, −0.05] |
| Old | 70 | Muslim error r | −0.34 [−0.53, −0.12] |

### 3.1.3. Stereotypes and Preferred Immigration Policy

How do stereotypes relate to immigration policy preferences? As far as we know, Carl [5] is the only study to have examined the relationship between immigration policy preferences for different origin country groups and the actual socioeconomic outcomes of those groups within their host country. He found evidence consistent with 'rational' policy preferences: Britons were more opposed to immigration of groups with higher arrest rates in the UK (the correlation between the log per capita arrest rate and net opposition was r = 0.69). His finding, along with the strong evidence of stereotype accuracy for immigrant groups and social groups in general suggests (but does not directly imply) that people's immigration policy preferences are based, at least in part, on reasonably accurate stereotypes about different origin country groups. This hypothesis was tested in the present study by fitting the path model shown in Figure 3, and by examining the correlation matrix shown in Table 3. (Note that one should not interpret the correlation between net fiscal contribution and Muslim population (%) as necessarily reflecting a causal effect of being Muslim on an individual's net fiscal contribution. For example, it is likely that being Muslim is related to variables like education, employment and number of children etc., and these other variables are the true causes of an individual's net fiscal contribution.)

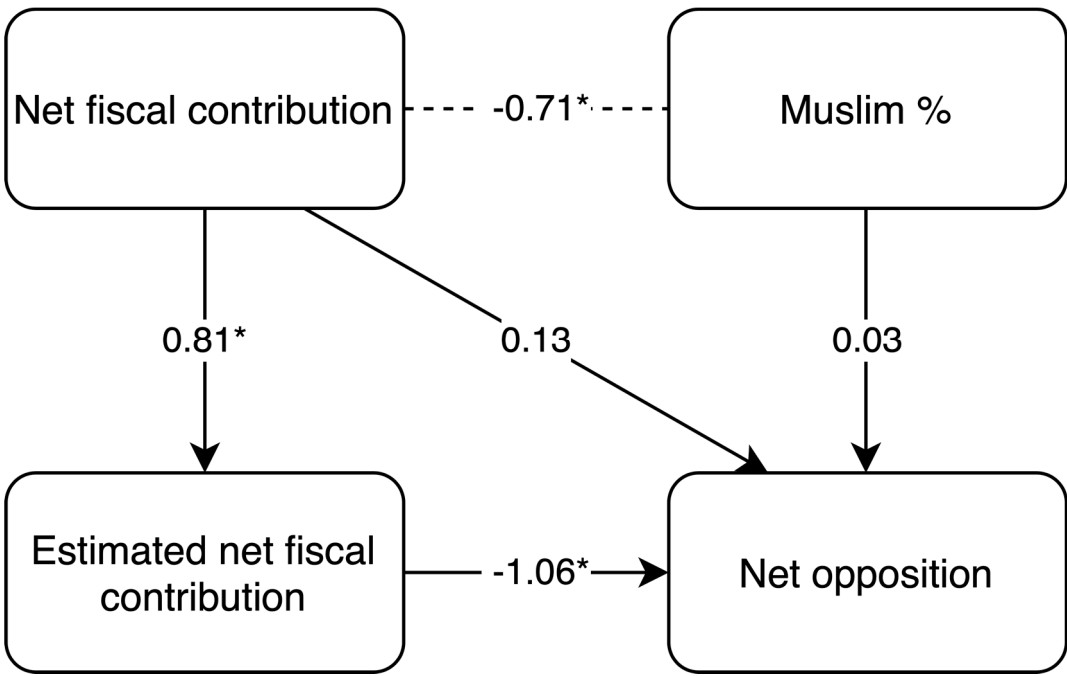

**Figure 3.** Path model for aggregate preferences. Values are standardized regression coefficients. Asterisks denote significance at *p* > 0.05 (Syria excluded).

**Table 3.** Correlation matrix for the primary aggregate-level variables. 'Net fiscal contribution' refers to each origin country group's actual net fiscal contribution in Denmark in 2014, according to the Danish Ministry of Finance. 'Mean estimate' refers to our respondents' mean estimate of each group's net fiscal contribution. 'Net opposition' refers to our respondents' net opposition to immigrants from each origin country. 'Muslim population (%)' refers to the percentage of Muslims in the home country.

|  | Net fiscal Contribution | Mean Estimate | Muslim Population (%) | Net Opposition |
|---|---|---|---|---|
| Net fiscal contribution | 1 |  |  |  |
| Mean estimate | 0.81 [0.64, 0.90] | 1 |  |  |
| Muslim population (%) | −0.73 [−0.86, −0.51] | −0.72 [−0.86, −0.50] | 1 |  |
| Net opposition | −0.75 [−0.87, −0.55] | −0.98 [−0.99, −0.96] | 0.70 [0.47, 0.85] | 1 |

Consistent with our expectation, respondents' stereotypes mediated a very large percentage of the relationship between actual net fiscal contribution and net opposition. In fact, the correlation between net opposition and estimated net fiscal contribution was r = −0.98. The correlation was also equal to r = −0.98 when excluding Syria (see Supplementary Materials). Figure 4 shows a scatterplot of the relationship between net opposition and estimated net fiscal contribution. There was a small direct effect of actual net fiscal contribution, but no discernible direct effect of percentage Muslim. Once we controlled for respondents' stereotypes, the strength of these variables' associations with net opposition decreased substantially. (Graphs showing estimates of net fiscal contributions and net opposition by age-group can be found in the Supplementary Materials.)

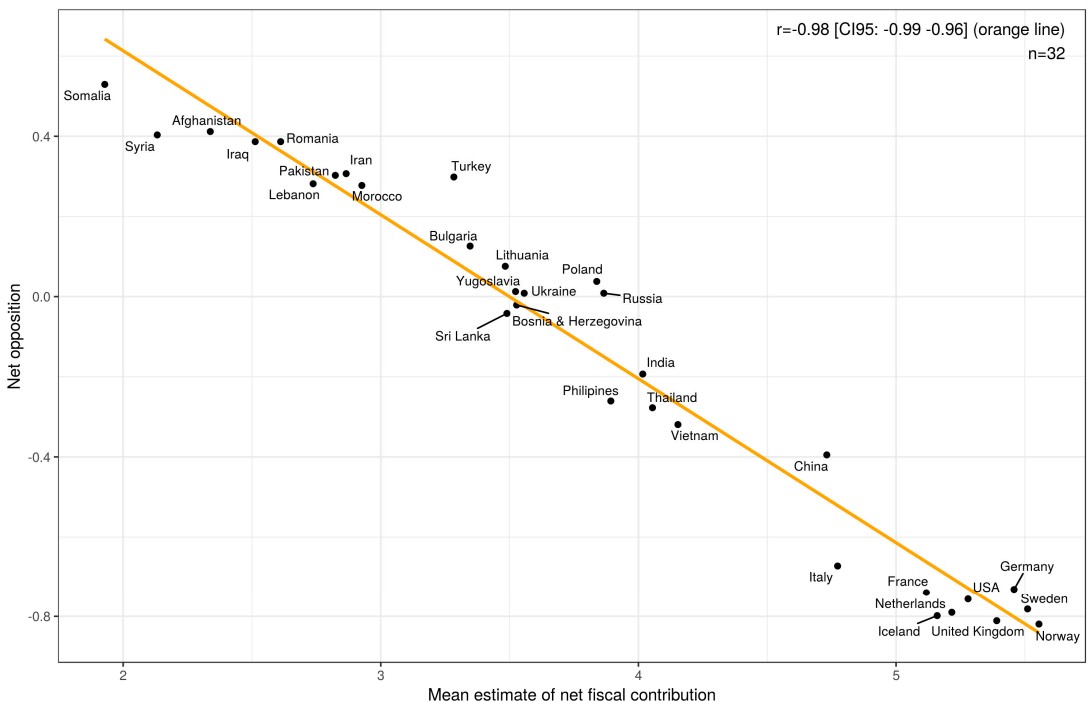

**Figure 4.** Scatterplot of net opposition and estimated net fiscal contribution (Pearson r).

In the pre-registration, we predicted a correlation of r = −0.77 (0.52, 0.90) between actual net fiscal contribution and net opposition. (Note that the actual correlation obtained by Carl [5] was r = 0.69, as noted in Section 1 of this paper. The reason our pre-registered value was r = 0.77, and not r = 0.69, is that it was based on the original version of Carl [5], which reported slightly different values for some of the analyses (see footnote 2 in that paper). These values were later corrected after a data entry error was discovered.) The observed value of r = 0.75 is therefore closely in line with what we had expected. This correlation increased to r = 0.82 when including Syria (see Supplementary Materials). Similarly, we predicted at least 50% mediation and we found essentially 100%. This was done using

the mediation package for R [34]. (Note that the *p*-value for the direct effect of net fiscal contribution on net opposition in the path model was greater than 0.05 for some estimation methods and less than 0.05 for others.) Thus, our prediction here was too conservative.

### 3.1.4. Comparison with Results from the Study of the United Kingdom

If stereotypes about origin country groups reflect underlying reality, they should be similar across countries because groups that have favorable outcomes in one European country tend to have favorable outcomes in other European countries as well [16,35–38].

Table 4 shows the correlation matrix between the primary variables across datasets.

**Table 4.** Primary variables across Danish and British datasets. DK = Denmark. UK = United Kingdom. 'Benefits use' is the proportion of people aged 30–39 who were receiving social benefits in Denmark in 2012. 'Net fiscal contribution' refers to each origin country group's net fiscal contribution in Denmark in 2014, according to the Danish Ministry of Finance. 'Crime rate' is the crime rate (see text for calculation), taken from Carl [5]. 'Mean estimate benefits' is the mean estimate of benefit use (as defined above), taken from Kirkegaard and Bjerrekær [6]. 'Mean estimate fiscal' is the mean estimate of net fiscal contribution from the present study. 'Net opposition' is the average net opposition, taken from Carl [5] and from the present study. 'Muslim population (%)' is from Pew Research's 2010 estimates, augmented with estimates for a few missing countries.

| | DK: Benefits Ue | DK: Net Fiscal Contribution | UK: Crime Rate | DK: Mean Estimate Benefits | DK: Mean Estimate Fiscal | DK: Net Opposition | UK: Net Opposition | Muslim Population (%) |
|---|---|---|---|---|---|---|---|---|
| DK: benefits use | 1 | | | | | | | |
| DK: net fiscal contribution | −0.89 | 1 | | | | | | |
| UK: crime rate | 0.51 | −0.41 | 1 | | | | | |
| DK: mean estimate benefits | 0.70 | −0.89 | 0.70 | 1 | | | | |
| DK: mean estimate fiscal | −0.72 | 0.81 | −0.70 | −0.94 | 1 | | | |
| DK: Net opposition | 0.68 | −0.75 | 0.73 | 0.90 | −0.98 | 1 | | |
| UK: Net opposition | 0.60 | −0.85 | 0.68 | 0.85 | −0.95 | 0.97 | 1 | |
| Muslim population (%) | 0.68 | −0.73 | 0.42 | 0.70 | −0.72 | 0.70 | 0.58 | 1 |

There are several results worth noting here. First, net opposition was nearly perfectly correlated across the two countries: r = 0.97 (although n = 12). This may be due, at least in part, to people relying on the same proxies for estimating the contribution of each group. For example, there is some evidence that people rely on the home country's GDP per capita: countries that are outliers for how well GDP per capita predicts the group's performance tend to be outliers in the same direction for stereotypes [6]. Second, there were moderate correlations for performance across countries, with crime rate in the UK correlating at about r = |0.41–0.51| with performance in Denmark. For the calculation of the UK crime rate, we used the average of the Z-scored log incarceration rate and log arrest rate. For more about the variables, see the original study (including the post-publication supplement). The comparatively low level of agreement is likely due to a noisy crime measure, as sampling variability has a large impact on the reliability of crime rates. Third, generally speaking, all the variables were strongly correlated and in a consistent fashion. The same groups tended to perform well in different countries, had similar stereotypes about them, and faced similar levels of opposition from the host population. The high degree of association between the variables is suggestive of common cause(s).

Note that the preceding analyses were not pre-registered because they were conceived after the data collection was completed.

### 3.2. Individual Stereotypes

### 3.2.1. Accuracy

Figure 5 shows the distribution of individual stereotype accuracy (Pearson's r).

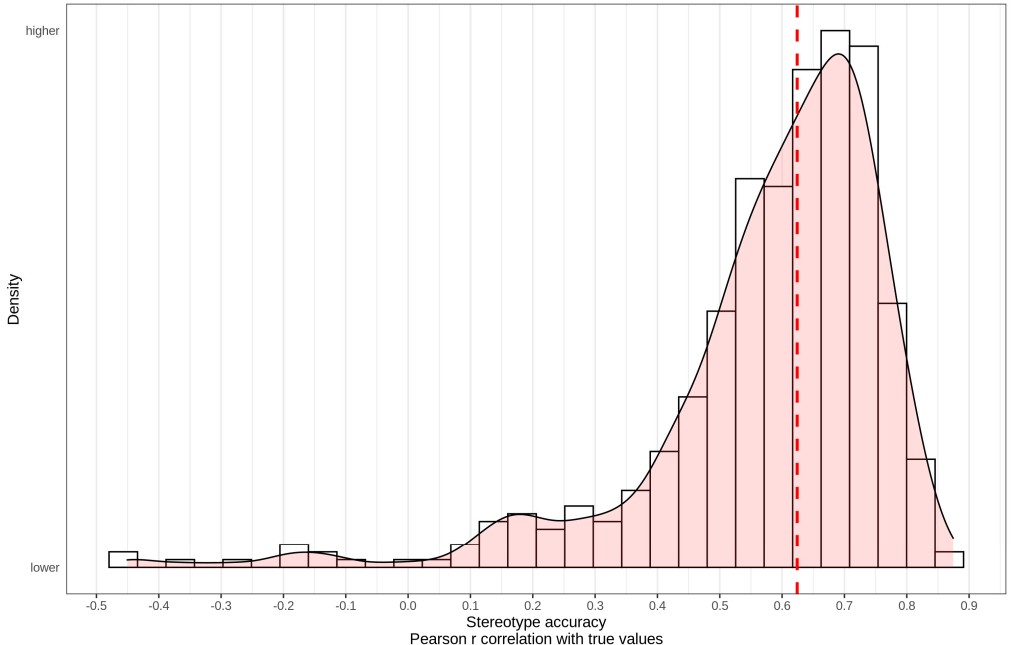

**Figure 5.** Distribution of individual stereotype accuracy (Pearson r). Vertical line = median.

As previously found, the distribution was highly skewed with most participants showing substantial accuracy, and a small subset of participants showing negative accuracy. The mean/median accuracy was r = 0.58/0.62. Jussim et al. [9] suggested using cutoffs of 0.30 and 0.50 for whether stereotypes can be considered accurate or not. Using these cutoffs, 91% and 78% of the stereotypes were found to be accurate. Median accuracy increased to r = 0.65 when excluding Syria (see Supplementary Materials).

### 3.2.2. Muslim Bias in Stereotypes

Stereotype bias metrics can be calculated the same way for individual stereotypes as for aggregate ones. Figure 6 shows the distribution of Muslim bias in stereotypes (i.e., the 'Muslim resid r' metric).

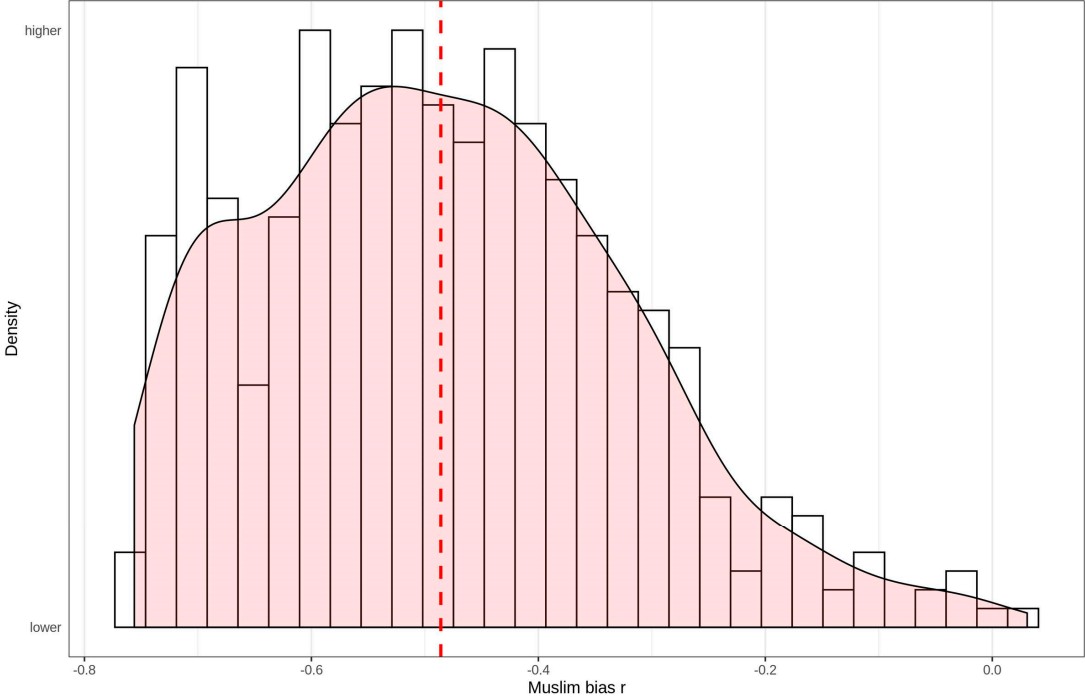

**Figure 6.** Distribution of Muslim bias (Muslim x residual r) in individual stereotypes. Vertical line = median.

In contrast to the aggregate data, the analysis of the individual level data found substantial evidence of biased estimates. The mean/median bias was r = −0.49/−0.49, with a range from r = −0.76 to 0.03. Only two persons out of 476 had a bias coefficient above zero, meaning that essentially all respondents (99.58%) tended to overestimate the net fiscal contribution of the groups with a higher percentage of Muslims in the home country.

### 3.2.3. Muslim Preferences

Just as one can examine mediation of immigrant preferences at the aggregate level, one can do the same at the individual level, though not necessarily with the same results. An analytical problem with examining mediation at the individual level is that our measures are not continuous, and may therefore be subject to measurement error related to the discretization [39,40]. The result is that mediation will appear to be smaller than it is, and residual variance will be attributed to other variables in so far as they correlate with the outcome. We used two approaches to examine mediation. First, we utilized ordinary least squares regression (OLS)/correlation. This was done by modeling immigration policy preferences as a function of estimated contribution, and then correlating the residuals with percentage Muslim. This correlation can be thought of as a measure of how strongly the participant prefers (or opposes) Muslim immigrants, holding their stereotypes constant. Second, we calculated the latent correlation [41] between policy preferences and estimated contributions in order to get an idea of how much downwards bias was present due to the use of non-continuous scales.

Using the OLS approach, we observed substantial evidence of mediation, with mean/median correlations of r = 0.57/0.67. When latent correlations were calculated, values increased to $r_{latent}$ = 0.63/0.78. The distribution of Muslim preference was close to normal with a mean/median of −0.11/−0.13. Negative values indicate that people tend to oppose Muslim immigrants, holding their stereotypes constant. However, given the issues with measurement error arising from non-continuous scales, the slight negative central tendency may have resulted from an imperfect control for stereotypes. We therefore utilized an alternative, non-pre-specified approach to examine mediation. Specifically, we calculated the (latent) correlations between preferences, stereotypes and percentage Muslim. We then used the correlation matrix to calculate the partial correlation between policy preferences and percentage Muslim, holding stereotypes constant. (This was done using *partial.r* from the psych package [42].) The results of this approach were in line with the OLS ones (r = 0.83): the mean/median partial correlation was −0.21/−0.27. Thus, respondents did oppose Muslim immigrants slightly more than one would expect based on what they estimated their net fiscal contributions to be.

### 3.2.4. Predictors of Stereotype Accuracy, Muslim Bias and Muslim Preference

We examined a number of sociodemographic correlates of the primary variables of interest: stereotype accuracy, Muslim bias in stereotypes, and Muslim preferences (for and against) with respect to immigration policy (using the partial correlation values). Table 5 displays the correlation matrix.

**Table 5.** Correlation matrix between the primary individual-level variables and predictors.

| | Stereotype Accuracy | Muslim Bias r | Muslim Preference | Age | Muslims are Treated Well | Admit Only Net Positive Immigrants | Non-Westerns are Net Positive |
|---|---|---|---|---|---|---|---|
| Stereotype accuracy | 1 | | | | | | |
| Muslim bias r | 0.74 | 1 | | | | | |
| Muslim preference | −0.04 | −0.05 | 1 | | | | |
| Age | 0.14 | 0.09 | −0.12 | 1 | | | |
| Muslims are treated well | 0.09 | 0.10 | −0.19 | 0.13 | 1 | | |
| Admit only net positive immigrants | 0.15 | 0.18 | −0.32 | 0.10 | 0.49 | 1 | |
| Non-westerns are net positive | −0.12 | −0.16 | 0.09 | −0.11 | −0.19 | −0.21 | 1 |

The positive correlation between Muslim bias and accuracy may seem odd until one recalls that nearly all respondents had a negative bias, meaning that higher values of bias connote less rather than more bias. This can be seen clearly in the scatterplot, shown in Figure 7. Specifically, it can be seen that the higher the accuracy, the lower the Muslim bias.

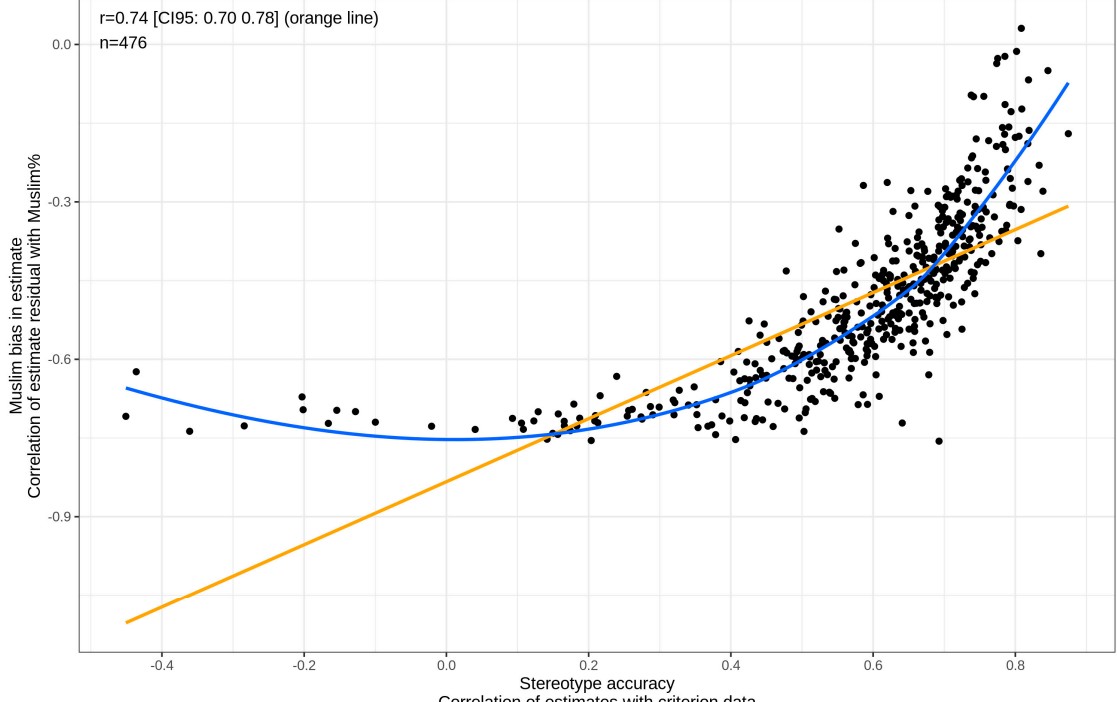

**Figure 7.** Scatterplot of stereotype accuracy (Pearson r) and Muslim bias (positive means anti-Muslim; negative means pro-Muslim). Orange line = linear fit. Blue line = local regression fit.

Stereotype accuracy was weakly (r's = 0.09, 0.15 and −0.12) associated with more restrictionist views on immigration, and Muslim bias showed the same pattern (r's = 0.10, 0.18 and −0.16). Our prediction for the Muslim bias x policy preferences correlation was r = |0.15| and we found an average of r = 0.15, meaning that the findings were in line with our numerical predictions. The Muslim preference metric showed the expected relationships to the policy preference questions, though they were fairly weak (r's = −0.19, −0.32, 0.09).

In a series of regression models with stereotype accuracy as the outcome, the combined effects of participant age, gender, education and party vote were examined. The results are displayed in Table 6. Note that we did not specify in our pre-registration how the party vote variable would be operationalized. Hence, we coded it in two different ways: first, we assigned parties to either the left-wing block or the right-wing block (Models 3 and 4); and second, we treated each party as a separate category (Models 5 and 6).

We were unable to explain much of the variance in stereotype accuracy: the adjusted $R^2$ ranged from 0.030 to 0.066. Age and education had consistent positive effects across the models (older people being more accurate, and those with higher levels of education being more accurate). Males were consistently more accurate than females, although the coefficient on gender was not always statistically significant. Support for the left-wing block was associated with slightly lower stereotype accuracy, although the coefficient was only significant in one of the two models. Specific party vote had no discernible predictive validity (each level of the categorical variable had $p > 0.05$, and the adjusted $R^2$ only increased from 0.030 to 0.038 as a result of its inclusion).

**Table 6.** Estimates from individual-level regression models. The outcome variable is the stereotype accuracy correlation. Entries are standardized betas. Note that we were unable to obtain education data for all respondents, meaning that the sample size for the models containing that variable was correspondingly reduced. The reference category for gender is 'male'; the reference category for block vote is 'right-wing block'; the reference category for party vote is 'Socialdemokratiet'. Significance levels: * 5%, ** 1%, *** 0.5%.

| | Model 1 | Model 2 | Model 3 | Model 4 | Model 5 | Model 6 |
|---|---|---|---|---|---|---|
| Intercept | 0.12 (0.063) | 0.20 ** (0.071) | 0.22 *** (0.078) | 0.29 *** (0.086) | 0.10 (0.119) | 0.21 (0.133) |
| Age | 0.13 *** (0.045) | 0.13 ** (0.051) | 0.13** (0.045) | 0.13 * (0.051) | 0.15 *** (0.047) | 0.12 * (0.054) |
| Female | −0.24 ** (0.091) | −0.19 (0.098) | −0.21* (0.092) | −0.15 (0.102) | −0.19 * (0.093) | −0.16 (0.105) |
| Education | | 0.15 *** (0.049) | | 0.16 *** (0.049) | | 0.16 *** (0.053) |
| Left-wing block | | | −0.20 * (0.098) | −0.20 (0.108) | | |
| Vote blank | | | −0.24 (0.164) | −0.22 (0.173) | | |
| Would not vote | | | −0.23 (0.218) | −0.10 (0.240) | | |
| Alternativet | | | | | −0.13 (0.217) | −0.27 (0.257) |
| Dansk Folkeparti | | | | | −0.08 (0.154) | 0.09 (0.168) |
| Enhedslisten | | | | | −0.11 (0.198) | −0.19 (0.204) |
| Konservative Folkeparti | | | | | 0.43 (0.317) | 0.03 (0.486) |
| Kristendemokraterne | | | | | 0.37 (0.416) | 0.19 (0.384) |
| Liberal Alliance | | | | | 0.37 (0.218) | 0.23 (0.248) |
| Nye Borgerlige | | | | | −0.09 (0.241) | −0.10 (0.265) |
| Radikale Venstre | | | | | −0.05 (0.215) | −0.28 (0.232) |
| Socialistisk Folkeparti | | | | | −0.25 (0.205) | 0.00 (0.239) |
| Venstre | | | | | 0.27 (0.176) | 0.07 (0.193) |
| Vote blank | | | | | −0.11 (0.186) | −0.14 (0.196) |
| Would not vote | | | | | −0.11 (0.234) | −0.02 (0.257) |
| $R^2$ adj. | 0.030 | 0.063 | 0.035 | 0.066 | 0.038 | 0.048 |
| N | 476 | 276 | 476 | 276 | 476 | 276 |

Both support for left- versus right-wing block and specific party vote were examined as sole predictors of Muslim bias and preference. The theoretical predictions here were that people supportive of right-wing and anti-immigration parties should display higher levels of bias against Muslims (i.e., lower levels of bias toward Muslims), and should display lower residual preferences toward them. These predictions were confirmed (see Supplementary Materials). Compared to individuals who support the left-wing block, those who support the right-wing block showed lower levels of bias ($d = -0.40$, $p < 0.001$), and lower residual preferences ($d = 0.24$, $p = 0.032$) toward Muslims. Differences between supporters of specific parties were in the expected directions, although many of the comparisons did not reach statistical significance. Interestingly, the average Muslim bias for every party was negative (i.e., there was a bias towards, rather than against, Muslims).

Finally, we examined whether the order of presentation had any effect on the main outcomes. No such effect was found (all model $R^2$ adjusted $\approx 0$, $p$'s $> 0.05$).

## 4. Discussion and Conclusions

This study has reported several findings of interest. First, the accuracy of stereotypes at both the aggregate and individual level was high, even compared to some previous results [6]. An accuracy correlation of $r = 0.81$ ($r = 0.85$ when excluding Syria) is quite remarkable and is large even for aggregate-level data (though we are not aware of any formal effect size analyses or guidelines). Similarly, the median individual accuracy observed of $r = 0.62$ ($r = 0.65$ when excluding Syria) is likewise larger than the majority of effects observed in social science [43,44].

Second, a slight (and sometimes non-significant) bias in the stereotypes with respect to Muslim origin country was observed at both the individual and aggregate levels. This bias, however, was in the opposite direction of what one would conventionally expect, i.e., stereotypes were found to be biased towards the Muslim-majority groups rather than against them. The effect size of the bias was similar to that found in a previous study [6], and is thus not dependent on the specific outcome stereotyped or the level of measurement (ratio vs. interval). One somewhat unexpected finding was that the bias was smaller for the aggregate stereotypes ($r = -0.25$) than for the median of the individual stereotypes ($r = -0.49$). This may be attributable to the use of ordinal-level data. An alternative interpretation is that it is due to the statistical confounding of percentage Muslim with low net fiscal contribution ($r = -0.73$, see Table 3). As individual-level stereotypes tend to be less accurate than aggregate-level stereotypes, one would expect below-average cases to be overestimated and likewise, for above-average cases to be underestimated. This is another way of saying that when the correlation is lower, the regression slope is further below one. Such confounding can be avoided by measuring stereotypes for an unrepresentative subsample of the countries where there is no relationship between percentage Muslim and socio-economic outcomes, or by using data from a different country where no such pattern exists. In general, this analytic problem represents one cost of working with a lower level of data, i.e., ordinal- and interval-level instead of ratio-level.

Third, at the aggregate level, stereotypes were found to nearly perfectly ($r = 0.98$, before and after excluding Syria) mediate the relationship between participants' immigration policy preferences and the criterion values. There was no evidence that percentage of Muslims in the home country affected people's preferences once stereotypes about fiscal contributions had been taken into account, in contrast to some previous findings [1,4].

At the individual-level, the evidence was less clear-cut. This was probably because the use of non-continuous data introduced measurement error related to discretiziation [45], which made the mediation analysis more difficult [40]. However, using two different analytic approaches, we found that participants were slightly more opposed (median $r$'s = $-0.13$ and $-0.27$) to groups with a higher percentage of Muslims in the home country once their stereotypes about fiscal contributions had been taken into account. This may reflect residual measurement error in the stereotypes (which results in predictive validity being absorbed by correlates; see Cole & Preacher [46]), preferences against Muslims related to other criteria (e.g., crime, attitudes towards women), religious animus, or general out-group

animus. It should be noted, however, that the pro-Muslim bias in stereotypes is in the opposite direction of the anti-Muslim opposition observed in participants' immigration policy preferences, so the effects cancel one another out to some extent.

Fourth, we were unable to explain much of the variance in stereotype accuracy. In particular, stereotype accuracy was not strongly related to participant's age (r = 0.14, β's = 0.12 to 0.15), gender (β's = 0.16 to 0.24), education level (r = 0.18, β's = 0.15 to 0.16), political party (voting intentions) or the answers to three questions about policy preferences. This replicates earlier findings for correlates of stereotype accuracy [6,8]. Apparently, whatever predicts stereotype accuracy is either a larger number of small factors or some as-of-yet unexplored factor. One characteristic that was recently shown to predict rapid learning of social stereotypes (based on data contrived by the researchers) is cognitive ability [47]. However, our survey did not include a measure of this characteristic (education may serve as a proxy for cognitive ability, and our measure did show weak to moderate associations with accuracy). Another possibility is that our outcome measure was hard to predict because it was measured with low reliability. We are not aware of any study that has reported a test-retest reliability (at the individual-level) for stereotype estimates, or derived accuracy and bias metrics. Obtaining estimates of these quantities is an important task for future research.

Fifth, both Muslim bias in stereotypes and Muslim preferences were found to be weakly related to party voting intentions in the expected direction. Supporters of more right-wing parties averaged slightly weaker levels of pro-Muslim bias, though they were still slightly biased towards them. Similarly, lower pro-Muslim bias was weakly related to restrictionist views on immigration. This was also true for stereotype accuracy.

Our study has a number of strengths. First, the sample size was large compared to most research in social psychology (median is 73–178; [48]) and was pre-specified, thereby avoiding any stopping related p-hacking. Second, the study used a fairly representative sample in contrast to most research in social psychology, which relies on convenience samples of university students [49]. Third, the study used pre-specified, strict quality control measures to avoid data contamination from inattentive survey responding, although dishonest responding could not be ruled out (see also Kirkegaard & Bjerrekær, [6]). Fourth, most of the analyses were pre-registered, thereby avoiding suspicious model overfitting by garden of forking paths [50,51]. Fifth, the study employed a randomized order of presentation for the survey sections to obviate any order-related effects. None were detected, however, despite the relatively large sample size.

It should be noted that there were a couple of minor deviations from the analyses we specified in our pre-registration: we employed an alternative measure of anti-Muslim preference in Section 3.2.3, and we included a few exploratory comparisons using the British data in Section 3.1.4. These additional analyses were conceived after data collection was completed.

Our study has a number of limitations. First, the stereotypes were not measured in natural units but using a 7-point scale. This prevented the use of methods that require ratio scale data and may have confounded some analyses. Second, immigration policy preferences were likewise measured using a simple 4-point scale. This was done to maximize comparability with a previous study [5] at the cost of measurement error. Third, there was no control for potentially dishonest survey responding [52]. Fourth, it could be argued that at least one of the questions we put to our respondents was somewhat ambiguous. Recall that the question designed to assess anti-Muslim preference was, 'Overall, how are Muslims treated in Denmark in comparison to non-Muslims?', and that respondents answered on a 1–7 Likert scale from 'much better' to 'much worse'. We assumed that respondents with an anti-Muslim preference would be more likely to answer 'much better' insofar as they might be resentful of Muslims allegedly being treated better than non-Muslims. However, it is possible that some respondents with an anti-Muslim preference actually answered 'much worse' because they believed that Muslims are being treated worse than non-Muslims, and they supported this. Fifth, people with higher levels of education were somewhat overrepresented in our sample, while those aged over 65 were not included at all. In addition, we did not collect data on a number of potentially important socioeconomic characteristics,

such as migrant background, race/ethnicity and household income. For these reasons, the external validity of our analyses should be considered somewhat limited.

One final limitation is that the same sample of respondents was asked to estimate each group's net fiscal contribution *and* to say whether there should be more or fewer immigrants from each origin country. This may have led to a correlation arising between the two questions, at least in part, due to a psychological need for respondents to give consistent answers [53,54]. Although recall that the order of sections was randomized for each respondent, and the order of countries was randomized within each section). A more powerful design, perhaps, would have been to obtain stereotypes from one sample and immigration policy preferences from another sample. However, we decided to administer both questions to the same sample of respondents so that we could examine the individual-level association between stereotypes and preferences. (Having said that, when we divided the sample into two halves at random, and then correlated stereotypes and policy preferences within and between each of the two halves, we found that the within-correlations were not substantially greater than the between-correlations.)

Our overall conclusion is that immigration policy preferences appear to be partly based on accurate stereotypes, at least in the Danish population. One implication of this conclusion is that public beliefs about immigrants are more accurate than is often assumed (e.g., [55–57]). Indeed, it is frequently asserted that the public beliefs about immigrants are largely erroneous, given that Europeans typically overestimate the immigrant fraction of the population by 10–15 percentage points [10,56]. Yet building on previous research [5–9], our study indicates that the Danish public has remarkably accurate beliefs about the relative positions of different origin country groups. A second implication of our conclusion is that the Danish public's immigration policy preferences may be motivated, at least in part, by straightforward concerns about the impact of different origin country groups on Danish public finances. Of course, it is highly likely that their immigration policy preferences also reflect motivations such as racism, xenophobia or in-group preference [3]. Indeed, in Carl's [5] study, percentage white in the home country predicted net opposition to immigrants even after controlling for immigrant crime rates. Finally, it is important to note that the preceding inference is limited by the possibility that self-fulfilling prophecy accounts for the correlations we observed. As argued in the Introduction, however, we regard this possibility as somewhat remote.

**Supplementary Materials:** The following are available online at http://www.mdpi.com/2075-4698/10/2/29/s1, Figure S1–S10; Table S1. Complete data and R notebook files are available at: https://osf.io/zb7es/.

**Author Contributions:** Conceptualization: E.O.W.K.; data curation: E.O.W.K.; methodology: E.O.W.K., N.C., J.D.B.; project administration: E.O.W.K.; formal analysis: E.O.W.K.; visualization: E.O.W.K.; writing—original draft: E.O.W.K.; writing—review & editing: E.O.W.K., N.C., J.D.B. All authors have read and agreed to the published version of the manuscript.

**Funding:** This research received no external funding.

**Conflicts of Interest:** The authors declare no conflict of interest.

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
