# Peer review of "Are Danes’ Immigration Policy Preferences Based on Accurate Stereotypes?"

_societies, doi:10.3390/soc10020029_

Round 1
Reviewer 1 Report
This is an interesting paper that analyzes the relevance of stereotypes about immigrants’ fiscal contribution for policy preferences among the general population of Denmark. It is a timely topic that has the potential to contribute significantly to academic and societal debates. The unique data on views toward specific origin groups is very interesting, and the explicit reflection on how the findings relate to previous studies helps to position the paper in this field of literature. Moreover, it is very helpful that the authors make efforts to be transparent about their reasoning underlying various decisions in the empirical strategy. That being said, I have a number of questions.
Question 1:
The study draws on a sample of the Danish population. The representativeness of that sample is key, given that the authors do not control for any compositional factors in their analyses. The risk of confounders is thus high.
I am not convinced that the sample is representative of the Danish population. Most troubling, the authors state on page 4 that they aim to take a sample that is representative among others on educational attainment. Yet two sentences later, they state that they “(…) did not collect data on respondents’ educational attainment”. How do the authors know that the data reflect the general population in this regard? Given the critical importance of this variable for everything that the authors are interested in (views towards immigrants (including Muslims), accuracy of views about immigrants, policy preferences, the list goes on), I find this highly problematic. Since non-response is strongly and positively associated with levels of education, while highly educated individuals are presumably a minority in the research population, oversampling on highly educated individuals seems likely.
Other relevant characteristics are not even mentioned in the generalizability discussion. For instance, how about migrant background? Clearly this will play an important role. How about socio-economic status? While I recognize that there are limits to the data that can be collected, in this case, the meaningful interpretation of the results depends on external validity (again, because there are no controls in the models).
I even have questions about some of the characteristics that are discussed. For instance, the mean age is mentioned, but clearly this is only part of the story. How about the distribution? If the sample contains mostly very young and very old individuals, and the research population is normally distributed in terms of age, then the mean of age will be roughly the same in both cases, but the sample will reflect the population poorly.
I would like to see a table with an overview of measures of central tendency and dispersion for both the research population and sample on relevant characteristics. In my view, this is a pre-condition for interpreting any of the findings.
Question 2:
It is clear that the authors study three main components, namely (1) immigration policy preferences, (2) immigrant stereotypes (on fiscal contribution) and (3) the accuracy of those stereotypes. However, the third element is in my view not well embedded in the story, and in effect a completely separate issue from the relationship between stereotypes and policy preferences. This is apparent in two important ways.
First, the introduction is predominantly about the potential relationship between stereotypes and policy preferences (including the potential for reverse causality). Whether or not these stereotypes are accurate is mentioned from time to time, but is not formally positioned in the theoretical framework. Indeed, the reader is left wondering if accuracy conditions the relevance of stereotypes for policy preferences, not to mention how this works in the reverse causality scenario.
Second, the accuracy of stereotypes is not explicitly modelled in the analysis either. While the authors analyze descriptively the extent to which stereotypes are accurate, this is not subsequently connected to the main question at hand, namely the relationship between stereotypes and policy preferences. At the moment, the authors first establish that stereotypes tend to be more or less accurate, and then frame stereotypes as accurate stereotypes in all subsequent analyses. But a correlation of .81/.85 is not a correlation of 1, and so I think it is misleading to speak in terms of accurate stereotypes in the analysis. Perhaps more importantly, the correlation analysis between stereotypes and policy preferences says nothing about the role of the accuracy of those stereotypes.
I would like to invite the authors to clarify the role of ‘accuracy’ in their study.
Question 3:
In a similar vein to the previous point, I do not understand the role of the so-called ‘Muslim bias’ as it is currently pitched in the paper. The authors argue that this is important to study because ‘[e]vidence indicates that many Muslim-majority communities in Denmark tend to have quite poor socio-economic outcomes’. They further argue that ‘the most plausible mechanism by which stereotypes might hamper the social performance of [this] group is other members of society underestimating the true social performance of that group’. In other words, the authors theorize that statistical discrimination is at the heart of the (in this case religious) divide in the labour market. I have three comments on this.
First, the authors do not actually study statistical discrimination in the labour market, nor does their data allow them to do so. In my view, the pitch of the paper will improve significantly if the focus stays on policy preference rather than labour market integration. In other words, it is not important to study views on Muslims specifically because of potential determinants of their labour market outcomes, but rather because the effect of stereotypes on policy preferences might be conditioned by the religious affiliation of the migrant group in question.
Second, I find that not only the theoretical but also empirical discussion on Muslim bias veers away from the central question on the role of stereotypes on policy preferences. In section 2.2.3., the authors analyze Muslim opposition holding stereotypes constant. But that seems to be a completely different research question from the rest of the paper.
Third, I am not convinced that the specific questions about Muslims in the survey actually measure Muslim bias. The question on how Muslims are treated is in principle descriptive, not normative. If one answers they are treated better, that may reflect reality, or perceived reality, but neither tells me with any degree of certainty what an individual thinks about that state of affairs, let alone what someone thinks about Muslims more generally. The same is true for individuals who answer they are treated poorly. In other words, I find the validity of this concept questionable.
I think it would be much better to (1) focus more on the main research question at hand, and (2) measure the potential relevance of Muslims in the story indirectly. More specifically, I would focus on how the relevance of (accurate) stereotypes for policy preferences is conditioned or not by the religious affiliation of the immigrant group, and I would operationalize Muslims simply through the proportion of Muslims in each of the origin countries. That is, is the effect of (accurate) stereotypes for policy preferences stronger or weaker (or the same) for migrants from countries with more/less Muslims?
Question 4:
In my view, the concept of stereotypes is used very loosely, but it is measured narrowly. More specifically, the study focuses on stereotypes regarding the fiscal contribution of migrants. While that is relevant and interesting, this is clearly only one way in which migrants can contribute to host societies, and only one aspect of how migrants are perceived by the general population. While it is of course not feasible to broaden this out significantly, and also not necessary to answer the central research question, I do think that the authors could be more specific in their terminology, so that the concepts actually reflect what is being measured and studied. Speaking in terms of stereotypes in general, and pitching the conclusions and implications of the study as such, in my view does not align with the empirics.
Author Response
General comments
We would like to begin by thanking both reviewers for their helpful and instructive comments. Before we respond to each of their points in detail, allow us to outline the major changes we have made to the manuscript:
- Added more relevant citations in support of our theoretical and methodological claims (e.g., in Section 1 and Section 3.14).
- Provided more descriptive statistics for our sample, as well as more detailed information about how our sample compares to the general population.
- Included a new individual-level multiple regression analysis of stereotype accuracy, which includes age, gender, education and two measures of party vote.
- Expanded the Discussion (Section 4).
- Revised and moderated our language, where necessary.
Reviewer 1
This is an interesting paper that analyzes the relevance of stereotypes about immigrants’ fiscal contribution for policy preferences among the general population of Denmark. It is a timely topic that has the potential to contribute significantly to academic and societal debates. The unique data on views toward specific origin groups is very interesting, and the explicit reflection on how the findings relate to previous studies helps to position the paper in this field of literature. Moreover, it is very helpful that the authors make efforts to be transparent about their reasoning underlying various decisions in the empirical strategy. That being said, I have a number of questions.
Question 1:
The study draws on a sample of the Danish population. The representativeness of that sample is key, given that the authors do not control for any compositional factors in their analyses. The risk of confounders is thus high.
I am not convinced that the sample is representative of the Danish population. Most troubling, the authors state on page 4 that they aim to take a sample that is representative among others on educational attainment. Yet two sentences later, they state that they “(…) did not collect data on respondents’ educational attainment”. How do the authors know that the data reflect the general population in this regard? Given the critical importance of this variable for everything that the authors are interested in (views towards immigrants (including Muslims), accuracy of views about immigrants, policy preferences, the list goes on), I find this highly problematic. Since non-response is strongly and positively associated with levels of education, while highly educated individuals are presumably a minority in the research population, oversampling on highly educated individuals seems likely.
Other relevant characteristics are not even mentioned in the generalizability discussion. For instance, how about migrant background? Clearly this will play an important role. How about socio-economic status? While I recognize that there are limits to the data that can be collected, in this case, the meaningful interpretation of the results depends on external validity (again, because there are no controls in the models).
I even have questions about some of the characteristics that are discussed. For instance, the mean age is mentioned, but clearly this is only part of the story. How about the distribution? If the sample contains mostly very young and very old individuals, and the research population is normally distributed in terms of age, then the mean of age will be roughly the same in both cases, but the sample will reflect the population poorly.
I would like to see a table with an overview of measures of central tendency and dispersion for both the research population and sample on relevant characteristics. In my view, this is a pre-condition for interpreting any of the findings.
Many thanks again to the reviewer for reading and commenting on the manuscript. We were told by the pollster that the aim was to obtain a sample that was approximately representative of the Danish population. However, we fully appreciate the reviewer’s concern that representativeness with respect to education might not have been achieved, given that we did not collect data on this variable. (Incidentally, as we have noted in Section 2, the lack of data on education was due to an oversight on our part at the design stage.) We contacted the pollster to see if it was possible to get data on education for our respondents. Luckily, the pollster was able to provide such data for 60% of our sample, based on archival sources. Whilst it would obviously be ideal to have education data for 100% of our sample, this does give us some idea about representativeness with respect to education. The first paragraph of Section 2 now reads:
[…] The mean age in our sample was 39.3 (SD = 13), compared to an expected value of 41.4 (SD = 14) in the general population. The percentage male in our sample was 51.3%, compared to an expected value of 50.5% in the general population. Due to an oversight at the design stage, we did not collect data on respondents’ educational attainment. However, the pollster was subsequently able to provide such data for around 60% of our respondents, based on archival sources. When we compared the distribution in our sample to that in the general population, we found that people with higher levels of education were somewhat over-represented in our sample. For example, the percentage without a high school degree was 9.1% in our sample, compared to 20.0% in the general population; the percentage with no more than a high school degree was 14.1% in our sample, compared to 11.3% in the general population; and the percentage with a university degree was 14.1% in our sample, compared to 11.5% in the general population.
As the reviewer expected, individuals with higher levels of education were overrepresented in our sample. In accordance with this observation, we made two other changes to the paper. First, we changed the first sentence of our Abstract to, “Stereotypes about 32 country-of-origin groups were measured using an online survey of the adult, non-elderly Danish population”. In other words, we removed the phrase “approximately representative”, so as not to give the impression that our sample achieved a high level of representativeness with respect to education. Second, we added a fifth point to the limitations section of the Discussion:
[…] people with higher levels of education were somewhat overrepresented in our sample, while those aged over 65 were not included at all. In addition, we did not collect data on a number of potentially important socio-economic characteristics, such as migrant background, race/ethnicity and household income. For these reasons, the external validity of our analyses should be considered somewhat limited.
(Note that this point also addresses the reviewer’s concern about our lack of data on other socio-economic characteristics.)
Regarding the age distribution, we have provided the standard deviation of age in our sample and the general population (see the paragraph above). As noted in footnote 2, the latter was obtained from Statistics Denmark, along with the other reference figures. We would prefer not to add a table, due to the fact that our manuscript already includes a large number of tables and figures. But we can obviously do so if the reviewer believes it is important.
Question 2:
It is clear that the authors study three main components, namely (1) immigration policy preferences, (2) immigrant stereotypes (on fiscal contribution) and (3) the accuracy of those stereotypes. However, the third element is in my view not well embedded in the story, and in effect a completely separate issue from the relationship between stereotypes and policy preferences. This is apparent in two important ways.
First, the introduction is predominantly about the potential relationship between stereotypes and policy preferences (including the potential for reverse causality). Whether or not these stereotypes are accurate is mentioned from time to time, but is not formally positioned in the theoretical framework. Indeed, the reader is left wondering if accuracy conditions the relevance of stereotypes for policy preferences, not to mention how this works in the reverse causality scenario.
Second, the accuracy of stereotypes is not explicitly modelled in the analysis either. While the authors analyze descriptively the extent to which stereotypes are accurate, this is not subsequently connected to the main question at hand, namely the relationship between stereotypes and policy preferences. At the moment, the authors first establish that stereotypes tend to be more or less accurate, and then frame stereotypes as accurate stereotypes in all subsequent analyses. But a correlation of .81/.85 is not a correlation of 1, and so I think it is misleading to speak in terms of accurate stereotypes in the analysis. Perhaps more importantly, the correlation analysis between stereotypes and policy preferences says nothing about the role of the accuracy of those stereotypes.
I would like to invite the authors to clarify the role of ‘accuracy’ in their study.
We appreciate the reviewer’s concerns here. However, we believe that we have modelled accuracy appropriately in our analyses, and that we are justified in describing the stereotypes we measured as “accurate”, given our findings and the conventions within the literature.
As our Introduction begins by noting, it is possible that “people rely, at least in part, on unmeasured but accurate stereotypes about origin country groups to inform their immigration policy preferences”. There is already a large body of evidence demonstrating that stereotypes about various demographic groups are generally quite accurate (see evidence cited on p. 2). Our study builds upon this literature by presenting evidence that Danes’ immigration policy preferences are partly informed by such stereotypes; in particular, accurate stereotypes about origin country groups’ net fiscal contributions.
In the Introduction, we analyse the concept of stereotype accuracy by pointing out that stereotypes could be accurate either because individuals accurately appraise differences in outcomes across groups based on their knowledge and experience or because individuals stereotypes’ actually cause differences in outcomes via the process of self-fulfilling prophecy. (Note, we also provide a number of citations to more detailed treatments of stereotype accuracy.) Our third paragraph, which reads as follows, is the most important one:
[…] it is unclear why members of the host population would come to hold particular stereotypes if those stereotypes did not correspond in some way with reality. One possibility is that Europeans would be naturally biased against, say, non-white groups or non-Christian groups. However, the fact that some non-white, non-Christian groups often have among the most favorable outcomes suggests that this is unlikely to be the case (see Neve, 2017). For example, Japanese living in the UK had the lowest average arrest rate in Carl’s (2016) study, while Indians living in Denmark had among the lowest average welfare use rates in Kirkegaard & Bjerrekær’s (2016) study. Moreover, there is evidence that differential immigrant outcomes can often be explained, at least in part, by skills present upon arrival. In other words, origin country groups with better education tend to have more favorable outcomes in their host country (Borjas, 2016; Hendricks & Schoellman, 2017; Carl, 2017) […] Given that the vast majority of immigrants arrive after completing their education, the host population’s stereotypes cannot have caused the observed differences in education levels across origin country groups. A plausible causal pathway is therefore from skills present upon arrival, to differential immigrant outcomes, to the host population’s stereotypes.
Regarding the role of stereotype accuracy in our aggregate-level analyses, the most important figure is Figure 3, which shows that actual net fiscal contribution has a much smaller association with net opposition once stereotypes are statistically controlled. This constitutes tentative evidence for our conjecture that “people rely, at least in part, on unmeasured but accurate stereotypes about origin country groups to inform their immigration policy preferences”. In line with the reviewer’s concern, we have added some statements at the end of Section 1 which aim to clarify the relationship between stereotype accuracy and immigration policy preferences in our study:
Our key hypotheses were that: respondents would hold accurate stereotypes about origin country groups’ net fiscal contributions; that they would display greater opposition to origin country groups with more negative net fiscal contributions in Denmark; and that the association between net fiscal contributions and immigration policy preferences would be mediated by respondents’ stereotypes about origin country groups’ net fiscal contributions.
As to whether it could be considered misleading to describe a stereotype accuracy correlation of .81 as “accurate”, we would respectfully disagree. First, a correlation of .81 is larger than the vast majority of effect sizes in social science (see evidence cited on p. 19), including ones that researchers might describe as “strong”, “very strong”, “large” or “very large”. Second, our use of “accurate” is consistent with guidelines specifically devised for stereotype accuracy research. In a paper titled ‘The Accuracy of Demographic Stereotypes’, Jussim et al. (2018) note the following:
For correlational accuracy, Jussim (2012), drawing heavily on Rosenthal and Rubin’s (1982) binomial effect size display, Cohen’s (1988) standards for small, medium and large effect sizes, and common sense, suggested that a high degree of correspondence between a belief and reality is indicated by a correlation of .40 or higher.
Question 3:
In a similar vein to the previous point, I do not understand the role of the so-called ‘Muslim bias’ as it is currently pitched in the paper. The authors argue that this is important to study because ‘[e]vidence indicates that many Muslim-majority communities in Denmark tend to have quite poor socio-economic outcomes’. They further argue that ‘the most plausible mechanism by which stereotypes might hamper the social performance of [this] group is other members of society underestimating the true social performance of that group’. In other words, the authors theorize that statistical discrimination is at the heart of the (in this case religious) divide in the labour market. I have three comments on this.
First, the authors do not actually study statistical discrimination in the labour market, nor does their data allow them to do so. In my view, the pitch of the paper will improve significantly if the focus stays on policy preference rather than labour market integration. In other words, it is not important to study views on Muslims specifically because of potential determinants of their labour market outcomes, but rather because the effect of stereotypes on policy preferences might be conditioned by the religious affiliation of the migrant group in question.
Second, I find that not only the theoretical but also empirical discussion on Muslim bias veers away from the central question on the role of stereotypes on policy preferences. In section 2.2.3., the authors analyze Muslim opposition holding stereotypes constant. But that seems to be a completely different research question from the rest of the paper.
Third, I am not convinced that the specific questions about Muslims in the survey actually measure Muslim bias. The question on how Muslims are treated is in principle descriptive, not normative. If one answers they are treated better, that may reflect reality, or perceived reality, but neither tells me with any degree of certainty what an individual thinks about that state of affairs, let alone what someone thinks about Muslims more generally. The same is true for individuals who answer they are treated poorly. In other words, I find the validity of this concept questionable.
I think it would be much better to (1) focus more on the main research question at hand, and (2) measure the potential relevance of Muslims in the story indirectly. More specifically, I would focus on how the relevance of (accurate) stereotypes for policy preferences is conditioned or not by the religious affiliation of the immigrant group, and I would operationalize Muslims simply through the proportion of Muslims in each of the origin countries. That is, is the effect of (accurate) stereotypes for policy preferences stronger or weaker (or the same) for migrants from countries with more/less Muslims?
We appreciate the reviewer’s concerns here, and we would like to thank him/her for suggesting a way to potentially simplify and improve our paper. However, we would prefer not to implement the changes he/she is suggesting because our analyses of Muslim bias were pre-registered. As a consequence, changing these analyses or not reporting them would be inconsistent with best research practice. For purposes of clarification, we have added the following statement on p. 9: “We should note that our study does not seek to make any claims about the level of employment or other discrimination against Muslims in Denmark.”
Question 4:
In my view, the concept of stereotypes is used very loosely, but it is measured narrowly. More specifically, the study focuses on stereotypes regarding the fiscal contribution of migrants. While that is relevant and interesting, this is clearly only one way in which migrants can contribute to host societies, and only one aspect of how migrants are perceived by the general population. While it is of course not feasible to broaden this out significantly, and also not necessary to answer the central research question, I do think that the authors could be more specific in their terminology, so that the concepts actually reflect what is being measured and studied. Speaking in terms of stereotypes in general, and pitching the conclusions and implications of the study as such, in my view does not align with the empirics.
Our conceptualisation of ‘stereotypes’ is consistent with the previous literature on stereotype accuracy in psychology (see, Jussim, 2012; Jussim et al., 2015). For example, in Section 3.1, we note:
Stereotype accuracy is commonly examined at both the individual (personal) level and the aggregate (consensual) level (Jussim et al., 2015). Each of these levels is interesting for different reasons. Aggregate stereotypes are theoretically the most important for social performance as they reflect the overall perception of a group, and are thus likely to influence the average or most prevalent behavior toward the group by other members of society (e.g., hiring practices). Individual stereotypes, on the other hand, provide the opportunity to examine person-level correlates of stereotype accuracy and bias. A relatively large body of research indicates that stereotypes are quite accurate at the aggregate level for most kinds of groups, especially major demographic groups […]
To address the reviewer’s point about the difference between stereotypes in general and stereotypes about net fiscal contributions in particular, we have added the following caveat on p. 6:
It is important to note that the stereotypes measured in our study correspond to a single criterion variable, namely net fiscal contribution, meaning that our study focusses on just one of the many domains in which migrants may influence their host societies. (We use the term ‘stereotypes’ as a shorthand for ‘stereotypes about origin country groups’ net fiscal contributions’.)
Reviewer 2 Report
This manuscript makes an important contribution to the scholarship on attitudes toward immigration and builds on the existing scholarship in a manner addressed clearly in the introduction.
I am somewhat dubious that respondents has distinct opinions about the economic contributions of each of the 32 country-of-origin groups. I would be interested in some discussion about how much variation individual respondents provided across the 32 countries. The means that are reported may mask the lack of ranking of some of the countries of origin at the extreme of the likert scale. I would expect this more on teh question of opposition to migration from certain countries. To the degree that a significant share of respondents are ranking fiscal contribution or net opposition at 1, the mean is less telling.
The manuscript reports that the questionnaire is available in the supplemental materials, but it was not available in the materials that I was able to access.
I was also interested in the decision to only collect data on respondents aged 18-65. Why were older respondents excluded? The political voices of the 66+ community are often felt disproportionately to their share of the population. I realize that it is not possible at this stage to correct the sample to include all adults, but I think that the author(s) should discuss the effects of the decision to focus on the 18-65 on the reported results. One test that could be run would be to look at the older respondents in the sample (55-65) to see if these older respondents differ from the full sample either on their estimates of each group's net fiscal contribution to Denmark or on attitudes toward who should be allowed to migrate to Denmark.
Author Response
General comments
We would like to begin by thanking both reviewers for their helpful and instructive comments. Before we respond to each of their points in detail, allow us to outline the major changes we have made to the manuscript:
- Added more relevant citations in support of our theoretical and methodological claims (e.g., in Section 1 and Section 3.14).
- Provided more descriptive statistics for our sample, as well as more detailed information about how our sample compares to the general population.
- Included a new individual-level multiple regression analysis of stereotype accuracy, which includes age, gender, education and two measures of party vote.
- Expanded the Discussion (Section 4).
- Revised and moderated our language, where necessary.
Reviewer 2
This manuscript makes an important contribution to the scholarship on attitudes toward immigration and builds on the existing scholarship in a manner addressed clearly in the introduction.
I am somewhat dubious that respondents has distinct opinions about the economic contributions of each of the 32 country-of-origin groups. I would be interested in some discussion about how much variation individual respondents provided across the 32 countries. The means that are reported may mask the lack of ranking of some of the countries of origin at the extreme of the likert scale. I would expect this more on teh question of opposition to migration from certain countries. To the degree that a significant share of respondents are ranking fiscal contribution or net opposition at 1, the mean is less telling.
Many thanks again to the reviewer for commenting on the manuscript. We computed full summary statistics for our respondents’ estimates of net fiscal contributions, as well as for their immigration policy preferences. We then correlated these with the actual net fiscal contributions for the various origin countries, taken from the Danish Ministry of Finance. In both cases, neither the median, the standard deviation, the skew, the kurtosis, nor the proportion of ‘1’ responses was a substantially better predictor of net fiscal contributions than the mean/net value. For example, the mean estimate of net fiscal contributions had a correlation of r = .81 with actual net fiscal contributions, whereas the median estimate had a correlation of r = .78, and the proportion of ‘1’ responses had a correlation r = -.83. Likewise, net opposition had a correlation of r = –.75 with net fiscal contributions, whereas the median estimate had a correlation of r = .60, and the proportion of ‘1’ responses had a correlation of r = .80. Although the proportion of ‘1’ responses had slightly higher correlations with net fiscal contributions than the mean/net values (to the tune of 2% and 5% of a standard deviation, respectively), we pre-registered the use of mean/net values. Given that the observed differences were very small, we would prefer to stick with our pre-registered plan.
The manuscript reports that the questionnaire is available in the supplemental materials, but it was not available in the materials that I was able to access.
We suspect that the Supporting Information was not provided by the journal subeditor in order to protect anonymity. We have attached a new version of the Supporting Information to our resubmission.
I was also interested in the decision to only collect data on respondents aged 18-65. Why were older respondents excluded? The political voices of the 66+ community are often felt disproportionately to their share of the population. I realize that it is not possible at this stage to correct the sample to include all adults, but I think that the author(s) should discuss the effects of the decision to focus on the 18-65 on the reported results. One test that could be run would be to look at the older respondents in the sample (55-65) to see if these older respondents differ from the full sample either on their estimates of each group's net fiscal contribution to Denmark or on attitudes toward who should be allowed to migrate to Denmark.
The decision not to collect data on those aged 66+ was made by the pollster, who could not be confident that such data would be representative. Note that we have provided additional information about the representativeness of the sample in Section 2. Specifically, the first paragraph of that section now reads as follows:
[…] The mean age in our sample was 39.3 (SD = 13), compared to an expected value of 41.4 (SD = 14) in the general population. The percentage male in our sample was 51.3%, compared to an expected value of 50.5% in the general population. Due to an oversight at the design stage, we did not collect data on respondents’ educational attainment. However, the pollster was subsequently able to provide such data for around 60% of our respondents, based on archival sources. When we compared the distribution in our sample to that in the general population, we found that people with higher levels of education were somewhat over-represented in our sample. For example, the percentage without a high school degree was 9.1% in our sample, compared to 20.0% in the general population; the percentage with no more than a high school degree was 14.1% in our sample, compared to 11.3% in the general population; and the percentage with a university degree was 14.1% in our sample, compared to 11.5% in the general population.
In addition, we have added a fifth point to the limitations section of the Discussion:
[…] people with higher levels of education were somewhat overrepresented in our sample, while those aged over 65 were not included at all. In addition, we did not collect data on a number of potentially important socio-economic characteristics, such as migrant background, race/ethnicity and household income. For these reasons, the external validity of our analyses should be considered somewhat limited.
Regarding the reviewer’s suggestion to see whether the respondents aged 55-65 gave systematically different responses from those aged 18-54, we partitioned our sample into three equal sized age-groups: those aged 18-31; those aged 31-46; and those aged 46-65. We then computed correlations between estimates of net fiscal contributions and actual net fiscal contributions, as well as between immigration policy preferences and net fiscal contributions, separately for each of the three age-groups. In both cases, the values for the three age-groups were very similar. In particular, the correlations between estimates of net fiscal contributions and actual net fiscal contributions for the three age-groups were: .82, .82 and .78. And the correlations between immigration policy preferences and net fiscal contributions were: .80, .75 and .71.
In addition, we computed the estimates of net fiscal contributions and net opposition values separately for the three age-groups. Estimates of net fiscal contributions were remarkably similar across the three age-groups, with almost no differences between them. Net opposition values followed the expected pattern (based on prior survey research), whereby older respondents generally showed greater opposition to immigration than younger respondents. This was particularly true for the countries for which the average level of net opposition was greatest. We have added two additional figures to our Supporting Information, which show estimates of net fiscal contributions and net opposition values separately for the three age-groups.
Round 2
Reviewer 1 Report
Dear authors,
Many thanks for the elaborate response to my comments. I am satisfied with these answers, and believe that the revisions have improved the paper. In particular, the extra data on education are really valuable in my view.